# Insight into the Biological Activity of Hennosides—Glucosides Isolated from *Lawsonia inermis* (henna): Could They Be Regarded as Active Constituents Instead

**DOI:** 10.3390/plants10020237

**Published:** 2021-01-26

**Authors:** Irina Maslovarić, Vesna Ilić, Ivana Drvenica, Ana Stančić, Slavko Mojsilović, Tamara Kukolj, Diana Bugarski, Luciano Saso, Marcello Nicoletti

**Affiliations:** 1Institute for Medical Research, University of Belgrade, Dr Subotića 4, POB 39, 11129 Belgrade 102, Serbia; irina.maslovaric@imi.bg.ac.rs (I.M.); vesnai@imi.bg.ac.rs (V.I.); ivana.drvenica@imi.bg.ac.rs (I.D.); ana_stancic@hotmail.com (A.S.); slavko@imi.bg.ac.rs (S.M.); tamara.kukolj@imi.bg.ac.rs (T.K.); dianab@imi.bg.ac.rs (D.B.); 2Department of Physiology and Pharmacology “Vittorio Erspamer”, Sapienza University of Rome, Square Aldo Moro, 5, 00185 Rome, Italy; luciano.saso@uniroma1.it; 3Department of Environmental Biology, Sapienza University of Rome, Square Aldo Moro, 5, 00185 Rome, Italy

**Keywords:** *Lawsonia inermis*, Lythraceae, glucosides, redox activity

## Abstract

Henna is the current name of the dye prepared from the dry leaf powder of *Lawsonia inermis* (Lythraceae). Several studies have focused on the chemistry and pharmacology of the henna dyeing active compound, lawsone, obtained from the main constituents of leaves, hennosides, during the processing of plant material. However, knowledge regarding the biological activity of hennosides is largely lacking. In this paper, the redox activity of three hennoside isomers is reported. The pro-oxidative activity was confirmed by their ability to induce mild lysis of erythrocytes and to increase the level of methemoglobin at the concentration ≥ 500 μg/mL. The antioxidant activity of hennosides (concentration ≥100 μg/mL) was determined by FRAP and ABTS assays. At concentration of 500 μg/mL, antioxidant activity of hennoside isomers was equivalent to 0.46 ± 0.08, 0.62 ± 0.28 and 0.35 ± 0.03 mM FeSO_4_ × 7H_2_O, and 0.15 ± 0.01, 0.30 ± 0.01 and 0.09 ± 0.01 mM Trolox. Hennosides at 100 μg/mL concentration did not influence viability of human breast cancer cell lines MDA231 and MCF-7 and primary human peripheral blood and periodontal ligament-mesenchymal stem cells, but produced a modest increase in concentration of antioxidants in the cell culture supernatants. The evidenced antioxidant and pro-oxidant activities indicate their potential to act as redox balance regulator, which opens up the possibility of using hennosides in commercial phytomedicines.

## 1. Introduction

*Lawsonia inermis* L. (syn. *L. alba* o *L. spinosa*, family Lythraceae), vulgarly known as henna, hennè, shudi, madurang, manghati, madayantika and goranti, is a perennial shrub, well known from ancient times as a medicinal and dyeing plant [1]. Nowadays, because of its strong coloring properties, the use of dried henna leaf powder for hair coloring and the body-decorating process, “temporary tattoos”, also known as mehndi [2,3], on the skin has increased. In the US, direct application to the skin is not approved by the Food and Drug Administration [4]. Conversely, it is authorized and widely available as a coloring agent for hair across the globe, including in the US [5]. The coloring action of red henna is due to lawsone, 2-hydroxy-1,4-naphtoquinone. Nonetheless, to obtain the dying effect of henna, the leaves must be treated until the principal active ingredient, lawsone, is able to react via Michael reaction with keratin in the skin and hair, resulting in a permanent dye stain:C_9_H_5_O_2_-C=O + _2_HN-Keratin → C_9_H_5_O_2_-C=N-Keratin + H_2_O

There is evidence, however, that lawsone, an aglycone, is an artifact arising from the oxidative transformation of the primary glycosidic constituents during the processing of plant material to obtain the dye [6]. The relation between glucosides and aglycone is common in natural products chemistry. Therefore, a preliminary treatment is necessary to convert the hennosides (glucosides), present originally in the raw material, into an active herbal dying principle, lawsone [7]. Conversion of hennosides into the unique aglycone requires that henna powder be soaked in mildly acidified solution for 6 to 24 h before use. Hennosides are three isomer glucosides, where each of the hydroxyls, derived from the interconversion of the two keto-enol forms of the naphtoquinone structure, can be glycosylated, leading to hennoside isomerism. The aglycone, derived from their hydrolysis, is then converted by oxidation into lawsone, the dying active compound (Figure 1) [7]. The henna drug content is usually referred to as the lawsone content, but as demonstrated [7], lawsone, as a free molecule, is absent in the raw material, and hennosides, precursors of henna, that are co-occurring in the activity with lawsone, could be regarded as active constituents instead [8] and serve to evaluate any biological activity of henna leaves exploited in traditional medicine in many countries.

In fact, although the cosmetic use of henna has obscured its medicinal importance, the use of the leaves is well present in all the cultures producing or using the plant. The Ebers Papyrus, dated to about 1550 B.C., is one of the oldest known scrolls and the main source of herbal knowledge of medicine in ancient Egypt [9]. Henna is reported as a pharmacologically important plant with significant in vitro and in vivo biological activities. Several pharmacological activities such as antibacterial, antioxidant, anti-inflammatory and anticancer properties have been documented, of which the antioxidant and antimicrobial activities have been most thoroughly investigated [1]. In the dried leaves, besides lawsone, other compounds present like, polyphenols and glucides are also co-occurring in the activity of henna [8]. However, the information about its most particular constituents is insufficient. In this paper, we evaluate the biological properties, redox, and cytotoxic activity of hennosides, designated A, B and C, respectively. Unlike lawsone, which is easily chemically produced, hennosides are difficult to obtain in pure form, and often they are referred to as lawsone. This study could be also of ethnopharmacological interest, to confirm or reconsider the use of henna in traditional medicine through the definition of hennoside activity, and it could serve as a valuable source of information for carrying out further studies on this plant in future.

## 2. Results

### 2.1. Antioxidant Activity of Hennosides

When hennosides were dissolved in 0.9% NaCl, both FRAP (ferric ion reducing antioxidant power) and ABTS (2,2′-azino-bis(3-ethylbenzothiazoline-6-sulfonic acid) assays showed antioxidant activity at hennoside concentration ≥ 100 μg/mL (Figure 2, Appendix A). The highest antioxidant activity was confirmed for hennoside B with both assays. When hennosides were dissolved in complete cell culture Dulbecco’s Modified Eagle Medium containing 10% fetal calf serum (D’MEM/10%FCS), their antioxidant capacity could not be measured with ABTS assay. This result demonstrated the high antioxidant capacity of the cell culture medium, indicating masking of antioxidant activity of hennosides by cell culture medium. The FRAP assay showed the antioxidant activity of hennosides in the cell culture media at hennoside concentration ≥ 100 μg/mL (Figure 2; Appendix A).

### 2.2. Effect of Hennosides on Erythrocyte lysis

The extent of hemolysis was determined by measuring the absorbance (optical density) at 540 nm (OD_540_) of supernatants obtained after centrifugation of erythrocyte suspension incubated with hennosides. A mild hemolysis was recorded in the presence of hennosides at ≥500 μg/mL concentration (Figure 3, Appendix A). Furthermore, the results showed the lack of hemolytic effect of dimethyl sulfoxide (DMSO), used as a stock solution for hennosides, in the concentrations applied.

The highest concentrations, ≥200 μg/mL, of hennosides in 0.9% NaCl changed the erythrocyte color from bright red to a dark red-violet color, which reflected the change in hemoglobin molecules. Some hennoside-induced hemoglobin structural changes are described in Section 2.3. Conversely, when erythrocytes and hennosides were diluted in 0.9% NaCl/10% FCS, the hennoside-induced erythrocyte color change was detected only at the concentration of 1000 μg/mL (Figure 4).

### 2.3. Effect of Hennosides on Hemoglobin

The pro-oxidative activity of hennosides was evaluated based on their capacity to modulate Vis spectral characteristics of outdated hemoglobin (Hb) in solution. The effect was dependent on hennoside isoform and solvent type used. VIS absorption spectroscopy has been employed as a universal method for investigating the structural changes of hemoglobin [10,11,12]. When 390–750 nm absorption spectrum of outdated Hb was analyzed, the absorption bands at 403–405, 541, 576 nm, characteristic of native, oxyhemoglobin, are identified. The porphyrin ring of the heme group showed an intensive absorption maximum at 403–407, i.e., the Soret band, and two absorption maxima at 541 and 577 nm (α and β) were derived from oxyhemoglobin. The absorbance at 630 nm originated from methemoglobin (oxidized, Fe^3+^ Hb), and indicates the partially oxidized hemoglobin (Figure 5, dashed line). Hennosides in concentration of 500 μg/mL showed changes towards further oxidation/denaturation of Hb. Markedly changed absorption spectrum can be noticed: (1) a blue spectral shift, toward shorter wavelength, of the Hb, (2) the decrease of Δα/Δβ value (indicates the transformation of Hb molecule to the oxidized form, (3) an increase in the ratio between the absorbance of Soret band and the absorbance at 577 (reflects the presence of free heme and the breakdown of Hb molecule), and (4) an increase in the absorbance value at 630 nm (indicates an increased methemoglobin level) (Figure 5, Table 1). The effect of hennosides was less pronounced in isotonic NaCl containing 10% FCS than in isotonic NaCl solution alone, establishing the protective role of the serum on Hb status.

The effect of hennosides on methemoglobin formation, i.e., an increase in absorbance (optical density) at 630 nm (OD_630_), was analyzed in a wide concentration range (1–1000 μg/mL) (Figure 6, Appendix A). In isotonic NaCl solution, the Hb OD_630_ values gradually increased at the concentration ≥ 500 μg/mL for each hennoside tested. In NaCl with 10% FCS the pronounced effect was detected only for hennoside A.

### 2.4. Effect of Hennosides on Cell Viability

After 24, 48 and 72 h, the viability of two breast cancer cell lines (MCF-7 and MDA 231) and two primary human mesenchymal stem cells (PDL-MSC and PB-MSC) was recorded by MTT (thiazolyl blue tetrazolium bromide) assay. Hennosides were used at a final concentration of 100 μg/mL. This concentration was chosen based on our experiment performed on erythrocytes and isolated Hb, where at this concentration of hennosides, neither erythrocyte lysis nor methemoglobin formation was assessed (Figure 3 and Figure 6). The concentration of 100 μg/mL was also the lowest concentration found to be responsible for antioxidant activity (Figure 2). The results showed a time-dependent change in the capacity of PB-MSC and MDA 231 cells, cultivated only in the cell culture medium, to reduce MTT. The enhancement in reduction of MTT was not detected in MCF-7 and PDL-MSC; the latter was compliant with our previous results [13] (Figure 7). We found no cytotoxicity for a given concentration of hennosides assessed (Table 2). Because of the uncertainty of measurement tool, 20% should be added or subtracted to any reading.

### 2.5. Antioxidant Activity of Culture Supernatant of Hennoside-Treated Cells

A moderate cell type- and incubation time-dependent increase in the concentration of antioxidants (antioxidant capacity) was detected in the culture supernatants of MDA 231 and MCF 7 breast cancer cell line and primary mesenchymal cells (PDL-MSC and PB-MSC) (Figure 8, Appendix A). The upward inclination in antioxidant concentration was not followed by the same trend of changes in MTT reactivity.

Analysis of breast cancer cell lines showed that a maximal increase in the antioxidants concentration was found in 24 h culture supernatants of hennoside A, B or C treated MDA 231 cells (36, 39 and 19% increase for hennoside A, B and C, respectively). A prolonged stimulatory effect was detected only for hennoside B, which increased the level of antioxidants in the 48 and 72 h MDA 231 cell culture supernatant by approximately 20%. The effect of hennosides on the concentration of antioxidants in supernatants of MCF 7 cells was weaker, and the stimulation did not exceed 25%, even in 24 h cultures.

The maximal stimulatory effect of hennoside A and B (an increase of 41% and 27%) on the antioxidants level in supernatants of PDL-MSC was detected in 48 h culture. The effect of hennoside A, but not hennoside B was still pronounced in 72 h cell culture (35% stimulation). Hennosides A and B also increased the antioxidant level in the supernatant of PB-MSC, it is just that stimulation, of approximately 30% and 40%, was pronounced in 24 and 48 h PB-MSC cultures. The effect of hennoside C on analyzed primary mesenchymal stem cells was weaker than the effect of hennoside A and B.

## 3. Discussion

From ancient Egypt until today, the use of henna has only expanded all over the world. Nowadays, the main use is cosmetic and for decorative body painting. However, henna possesses curative properties due to the presence of various complex chemical substances of different composition. Each of those chemicals that henna contains, in addition to the well-known lawsone, a widely used hair dye, can exert its own biological effect on the body [14]. Henna is considered safe. Lawsone’s position is slightly different: a few studies pointed out that lawsone carries the potential to induce oxidative injury to red blood cells (RBC) with normal Glucose-6-phosphate dehydrogenase (G6PD) activity, and even more so to G6PD-deficient red cells, and acute renal failure was observed due to its nephrotoxic effect [15,16].

To the best of our knowledge, there is a lack of data on the biological effects of hennosides, and we found it to be of interest to explore the redox potential of these isomers using erythrocytes, breast cancer cell lines and primary mesenchymal stem cells, as a model system.

In this study, we reported the ability of hennosides to provoke hemolytic response, and oxidation of hemoglobin to methemoglobin, although higher concentrations (≥500 μg/mL) of hennosides are necessary to achieve this pro-oxidative effect. Additionally, a mild hemolysis induced by hennosides dissolved in isotonic NaCl solution with serum addition was not followed by a change in the color of hemoglobin; the UV-Vis spectra of isolated hemoglobin indicated the protective effect of serum. To substantiate the protective effect of serum, we will quote Kameneva et al. [17]: “The lowest hemolysis was obtained for RBCs, human and bovine, resuspended in serum, plasma, and albumin solutions”.

From the results outlined above, we also witnessed the ability of hennosides to induce methemoglobin formation in vitro. Furthermore, the results confirmed the influence of solvent on methemoglobin formation i.e., the presence of serum in the solution dampened the oxidative “pressure” on Hb. The pro-oxidant behavior of hennosides finds its support in the observation presented by McMillan et al. [18] that lawsone (1,4-naphthoquinone) carries the potential to produce oxidant pressure and damage, to rat erythrocytes in vivo. They also reported that hemolytic concentration (0.5 mM) of two 1,4-naphthoquinone derivatives induced significant methemoglobin formation in rat erythrocytes in vitro. The in vitro hemolytic concentration of hennosides (1.4 mM) reported in our work is close to three times as high as in vivo hemolytic concentration of naphthoquinones (0.5 mM).

Moreover, we concluded, based on our results for cytotoxicity, that the assessed concentration (100 μg/mL) of hennoside-treated cells did not reduce the viability of either cancer or MSC cells. However, it is necessary to screen higher concentrations of hennosides for toxic effect on cells.

In the supernatants of hennoside-treated cancer and MSC cells, we confirmed the antioxidant protection at a 100 μg/mL concentration. Another step in evaluating the redox activity of hennosides dissolved in isotonic NaCl or D’MEM/10%FCS gave us an estimate that hennosides at 100 μg/mL and higher concentration can act as reducing agents. At this point, we need to emphasize that our estimation of antioxidant activity of hennosides in cell culture supernatants was based only on FRAP assay; ABTS assay has already proved to be inadequate for hennosides in the presence of cell culture medium enriched with serum. These differences may be due to various kinds of antioxidants present in the samples which react differently with the radicals used [19] and the known fact that medium supplemented with serum is expected to have appropriately higher antioxidant capacity [20].

As we mentioned earlier, the major constituent of henna plant preparations is lawsone that has mainly been used in the synthesis of a few anticancer drugs, along with henna’s chemical breakdown. Although many medical advantages are attributed to lawsone, its transport represents a challenge associated with its hydrophobic nature, which results in poor solubility, poor permeability, low bioavailability and instability in biological systems [21,22]. Conversely, glucosides, although chemically difficult to obtain, are marked by better solubility and delivery into the cell compared to active but unstable lawsone [7], which ultimately directed our interest towards hennosides as constituents.

Current direct information about the pharmacologic activities of hennosides are lacking, due to the difficulties in obtaining them as pure compounds and the misunderstanding regarding the lawsone being the active ingredient of henna. However, it is clear that there is a structural relation between hennosides and lawsone, and the reports on lawsone are indicative of the effects reported in this paper. The first indication concerns the application of henna, which can induce hemolytic anemia. However, phytochemical studies evidenced in henna the presence of hundreds of constituents, several of them specific to the plant. Lawsone is thought to be the causative agent [23], since its administration to rats has been shown to induce a hemolytic response associated with oxidative damage to erythrocytes. However, when isolated erythrocytes were directly exposed to lawsone, no provoked oxidative damage was evidenced, suggesting that lawsone must undergo extra-erythrocytic bioactivation in vivo. A study [18] reported several relevant data, suggesting that lawsone is a weak direct-acting hemolytic agent that does not require extra-erythrocytic metabolism to cause hemotoxicity. These considerations are based on the observation that neither lawsone nor its hydroquininic form (1,2,4-trihydroxynaphthalene) were directly hemolytic or methemoglobinemic, even at high concentrations. Furthermore, lawsone had no effect on erythrocytic GSH (glutathione) levels, whereas its hydroquinonic form induced a modest depletion (approximately 30%). Therefore, the conclusion of the study is that the hemolytic response to henna could be restricted to individuals with compromised antioxidant defense, as well as a confirmation of the association between the hemolytic effect and the oxidative damages. These data are in accordance with the use of dyeing plants in pigmentation and depigmentation products whose mechanism of action is based on melanin production and antioxidant effects. From a structural point of view, it is necessary to perceive lawsone as the result of equilibrium between the quinonic and the hydroquinonic form through the keto-enolic tautomerism, and it is considered that the antioxidant activity is attributed to the presence of functional phenolic and enolic groups. This structure/activity relation, confirmed by many studies in henna and L. inermis extracts [24], makes it possible to suggest a preference for plant extracts in cosmetics for body art, the treatment of skin causing often local inflammation and dermatitis [25]. In particular, structures of hennosides also possess a keto-enolic tautomerism, but the glucoside form changes radically solubility, biodisponibility and reactivity in favor of further in-deep analysis of the properties here reported.

The main outcome of the results obtained in this study is that hennoside anti- and pro-oxidant activities indicate their potential to act as redox balance regulator. It is known that in inflammation associated with infectious, autoimmune and malignant diseases, a common denominator of body response to stressors is the redox homeostasis; an equilibrium between the production of reactive oxidative species (ROS), signaling molecules that play important roles in cellular physiology and pathophysiology necessary for initiation of multiple defense mechanisms of the body, but can also damage cellular proteins, lipids, and DNA when in excess, and the antioxidant defense system important for resolution of inflammation [26,27,28]. Although it has for a long time been considered that ROS synthesis and ROS-induced oxidative damage of cells and tissues are harmful, today it is a known fact that their generation (even at low-to-moderate levels) represents an important signaling and protection mechanism in infectious and malignant diseases [25,26]. Whether hennoside isomers, as potential therapeutics, will act either as pro- or antioxidant agents depends on their concentration, the molecular substrates they react with, the nutritional status and immunological competence of individuals, as well as the inflammatory milieu in organs and tissues affected by the disease. Therefore, it is necessary to proceed with further investigation to determinate the mechanisms of action of hennosides, and consequently their possible pharmacological use, as well as the safe use of henna products in body art, like temporary tattoos and hair dye [29,30].

## 4. Materials and Methods

### 4.1. Plant Material, Authentication

Leaves from samples of *Lawsonia inermis* L. were collected in the Shahdad zone of Kerman (Iran) at the end of June, before the flowering period. To collect the leaves in time, it is necessary to follow the withdrawal dates of the Indian monsoon and its relation with the onset of fall precipitation in different parts of Iran. The collected leaves were immediately dried at ordinary temperature using an air flux.

Leaves were identified and authenticated by M. Nicoletti based on morpho-anatomical and histochemical characters. Voucher samples were deposited into the Herbarium at Department of Environmental Biology, Sapienza University of Rome.

### 4.2. Sample Preparation

The dried leaves (20 g) were crushed, powdered and sieved (sieve No. 355) to obtain a uniform powder. Later, the powder was extracted with methanol (100 mL) at ordinary temperature for 2 days and the material was transferred in capped test tubes, gently shaken and then sonicated in a model 3210E-MTH (Branson, Danbury, CT, USA) ultrasonic bath for 45 min [31,32]. The supernatant was filtered through Albet (Barcelona, Spain) qualitative analysis filter paper (43–38 μm) and the filtered extract was concentrated under vacuum to dryness to obtain solid amorphous residues. The yield of the obtained dried extract was about 18% (*w/w*). The obtained extract was separated by careful column chromatography on silica gel (400 g), collecting fractions of 10 mL. First, the column was eluted with 20 mL of n-hexane, and later with n-hexane/ethylacetate 99:1 (*w/w*), obtaining pure hennosides A–C in the fractions 15–25, with the following yields: hennoside A (10 mg), hennoside C (15 mg), hennoside B (8 mg), in the sequence. The total procedure was repeated three times to obtain the sufficient quantities of pure compounds. The hennosides were identified by their 1H NMR spectra (Varian 400 in CDCl3; Varian, Inc., Palo Alto, CA, USA) (Table 3), as already reported Nicoletti et al. [33].

### 4.3. Antioxidant Capacity Estimation

Hennosides were diluted in two different solvents (0.9% NaCl and D’MEM/10% FCS) in the following concentrations: 1, 10, 100, 200 500, 1000 μg/mL. The total antioxidant capacity was measured using ferric ion reducing antioxidant power (FRAP) and 2,2′-azino-bis(3-ethylbenzothiazoline-6-sulphonic acid (ABTS) de-colorization assay. Additionally, supernatants of the cells co-cultivated with 100 μg/mL hennosides were also analyzed.

#### 4.3.1. Ferric-Reducing Antioxidant Power Assay

FRAP was measured as described by Benzie and Strain [34], with some modifications. A working FRAP solution was prepared by mixing 10 mL 300 mM acetate buffer, pH 3.6; 1 mL 10 mM 2,4,6-Tripyridyl-s-triazine; 1 mL 20 mM FeCl_3_; 1.2 mL H_2_O. Undiluted samples (10 µL) containing 1, 10, 100, 200, 500 and 1000 μg/mL hennosides A, B or C dissolved in 0.9% NaCl or in D’MEM/10% FCS and cell culture supernatants of different cells grown with 100 μg/mL hennosides A, B or C in (see Section 4.7) were added to flat bottom microplates (96-well) together with working FRAP solution (300 µL) and incubated at 37 °C in the dark for 2 h before reading its absorbance at 540 nm using a multiplate reader (Victor2V 1420 multilabel HTS counter; Wallac, Finland). Absorbance range was 0.210–0.390. The increase in absorbance is proportional to the activity of antioxidants in the sample. A calibration curve was made of FeSO_4_ × 7H_2_O with a set of following concentrations (0.1, 0.2, 0.4, 0.6, 0.8, and 1.0 mM, and absorbance range 0.100–0.400) and the concentration of antioxidants was expressed as mM equivalent to FeSO_4_ × 7H_2_O. Each concentration of hennosides A, B and C was tested in triplicate.

#### 4.3.2. ABTS Radical Cation Decolorization Assay

ABTS radical scavenging activity was determined based on the method described by Miller et al. [35], with few modifications. Undiluted samples (10 µL) of 1, 10, 100, 200, 500, and 1000 μg/mL of hennosides A, B and C in 0.9% NaCl or 10 times diluted samples (10 µL) containing the same concentration of hennosides but diluted in D’MEM/10%FCS were mixed with 290 µL ABTS solution (ABTS•+ radical cation formed by chemical oxidation of 5 mL 1.98 mM ABTS with 88 µL 10.23 mM potassium peroxydisulfate). After incubation for 15 min at 37 °C, the absorbance of the mixture was measured in a multiplate reader (Victor2V 1420 multilabel HTS counter; Wallac, Finland) at 405 nm. Absorbance range was 0.100–0.250. The extent of ABTS•+ decrease in absorbance is proportional to the activity of antioxidants in each sample. A calibration curve was made of Trolox (6-hydroxy-2,5,7,8-tetramethylchroman-2-carboxylic acid), water soluble vitamin E analog, (0.05, 0.1, 0.2, 0.3, 0.4, 0.5 and 0.6 mM, and absorbance range 0.560–0.040) and the concentration of antioxidants was expressed as mM equivalent to Trolox. Each concentration of hennosides A, B and C was tested in triplicate.

### 4.4. In Vitro Erythrocyte Lysis

Human erythrocyte samples were obtained from the Institute for Transfusiology and Hemobiology, Military Medical Academy, Belgrade, Serbia. Erythrocytes were isolated from buffy coat on the same day of collection. Anonymous buffy coats were obtained from healthy volunteer blood donors, and the bridge to donor is broken. The cells were already labeled as waste.

Experiment was carried out by addition of hennosides A, B and C (200 μL) in the concentration range between 1 ng/mL–1 mg/mL, dissolved in 0.9% NaCl or 0.9% NaCl containing 10% FCS (NaCl-FCS), to 20 μL of erythrocytes, previously washed to remove the plasma and leukocytes, and re-suspended in 0.9% NaCl to an 8% hematocrit. The assay was performed in 96 well, U-bottom shaped, microtiter plate. Each concentration of hennoside A, B and C was tested in triplicate. For preparing the stock solution (5 mg/mL) each hennoside was dissolved in DMSO. To assess the hemolytic effect of DMSO on erythrocytes, the erythrocytes were incubated with corresponding dilution of DMSO. After the incubation time of 90 min at 37 °C, and centrifugation for 10 min at 2500 rpm (Megafuge 1.0R Heraeus, Landshut, Germany), OD_540_ value (absorbance at 540 nm; corresponds one of absorption maxima of oxyhemoglobin) of supernatant was measured in a multiplate reader (Victor2V 1420 multilable HTS counter; Wallac, Finland.

### 4.5. VIS Spectroscopy of Hemoglobin

The hennosides A, B and C in the concentration range between 1 ng/mL–1 mg/mL were incubated with outdated bovine hemoglobin (bHb) (erythrocytes derived from cattle slaughterhouse, as described by Drvenica et al. [36], dissolved in 0.9% NaCl or 0.9 NaCl containing 10% FCS (NaCl-FCS). After the incubation period of 90 min at 37 °C, the OD_630_ value, which is the methemoglobin characteristic absorption peak, was obtained on microplate reader. Effect of hennosides (0.5 mg/mL) on the absorption spectra (wavelength range between 390 and 750 nm) of bHb (10 μM) was analyzed with Ultrospec 3300 pro UV-VIS spectrophotometer (Amersham Biosciences, Little Chalfont, UK).

### 4.6. MTT (Thiazolyl Blue Tetrazolium Bromide) Assay

Human breast carcinoma cell lines MDA231 and MCF7 (ATCC, Manassas, VA, USA), human peripheral blood derived mesenchymal stem cells (PB-MSCs) [37] and human periodontal ligament derived mesenchymal stem cells (PDL-MSCs) [38] were cultured in D’MEM cell growth medium (Sigma-Aldrich, St. Louis, MO, USA) enriched with 10% foetal bovine serum (FCS) (PAA Laboratories GmbH, Pasching, Austria) (D’MEM-10% FCS). A passage number of the MSCs, used in the experiment, was less than 10.

Cells were seeded in 25 cm^2^ plastic tissue culture flasks at a concentration of 2 × 10^5^ cells/mL and incubated at 37 °C with humidified atmosphere of 5% CO_2_. The cells were maintained by replacing the growth medium every 2 to 3 days. After having reached 80–90% of confluence, the cells were detached by 0.25% trypsin-EDTA solution (PAA) and re-plated at a seeding density appropriate for the cell line.

MTT test was performed to analyze the viability of cells used in this study [39]. In brief, 1 × 10^4^ cells in D’MEM-10%FCS solution/well were seeded in 96-well plates and incubated for 24 h. After adhesion, the incubated cells were treated with hennoside A, B and C, respectively, and diluted in D’MEM/10%FCS to reach the final concentration of 100 μg/mL per well (the final volume per well was 100 µL). Cells in 100 µL D’MEM-10%FCS suspension was used as a control for the cell baseline proliferation level. The incubation was continued for 24, 48 and 72 h. After incubation, 10 µL of MTT (thiazolyl blue tetrazolium bromide, i.e., 3-(4,5-dimethylthiazol-2-yl)-2,5-diphenyltetrazolium bromide) (Sigma-Aldrich, USA) solution (5 mg/mL in PBS) was applied and incubation was carried out at 37 °C for 3 h in a dark place. The medium was then replaced with 100 µL of 10% SDS containing 10 mM HCl to dissolve the formazan crystals. The solubilized formazan dye product of each well was quantified spectrophotometrically at 540 nm with a 96-well plate reader and 620 nm was used as the reference wavelength. Blanks containing only culture medium, MTT and SDS were used to correct the absorbance. Relative MSC viability in D’MEM/10%FCS, as an index to MSC viability, was set at 100%.

### 4.7. Antioxidant Activity in the Cell Culture Supernatants

Human breast carcinoma cell lines MDA231 and MCF7, human peripheral blood-derived mesenchymal stem cells (PB-MSCs) were prepared and cultured with and without 100 μg/mL for 24, 48 and 72 h in 96 well microtiter plates, as described in the Section 4.6. After the culture time, microtiter plates were centrifuged (10 min, 2500 rpm, 22 °C), and supernatant was collected. The antioxidant activity of the cell culture supernatants was determined by FRAP assay as described in Section 4.3.1.

### 4.8. Statistical Analysis

Graphs were created with Microsoft Office Excel software (Microsoft Corporation, USA). In the tables in the Appendix A, the results are expressed as means with standard deviation (SD).

## 5. Conclusions

The results of this study delineated threes isomers of hennosides, the main constituents of *Lawsonia inermis* (henna tree), as redox active compounds. They were able to provoke hemolytic response, and oxidation of hemoglobin, although higher concentrations (≥500 μg/mL) of hennosides were necessary to achieve this pro-oxidative effect. Antioxidant activity of hennosides at concentration ≥ 100 μg/mL was confirmed by FRAP and ABTS assays. At 100 μg/mL concentration, hennosides did not influence the viability of human breast cancer cell lines MDA231 and MCF-7 and primary human peripheral blood- and periodontal ligament-mesenchymal stem cells, they induced a moderate increase in concentration of antioxidants in the cell culture supernatants instead.

Even though we have just scratched the surface of the biological activity of glucosides/hennosides, this research gives a first insight into the redox activity of hennoside extracts in vitro. Hennosides are examples of substances that have yet to be tested to ascertain what they can do and how they can regulate the redox potential of cells. It is noteworthy to quote Glasauer and Chandel [25] that “pro-oxidant cancer therapy makes an interesting area of study nowadays”.

## Figures and Tables

**Figure 1 plants-10-00237-f001:**
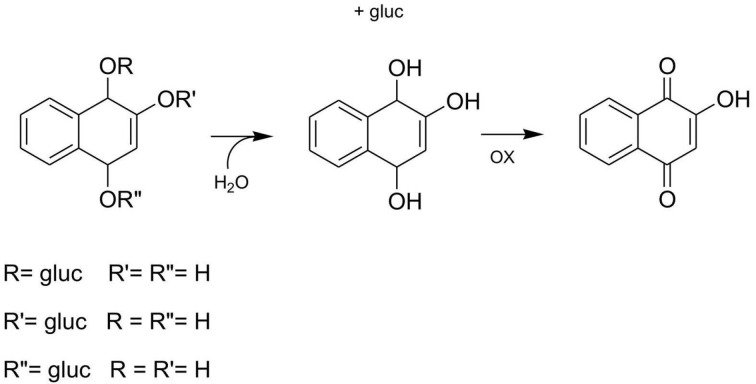
Conversion of hennosides (isomerglucosides) into lawsone (aglycone).

**Figure 2 plants-10-00237-f002:**
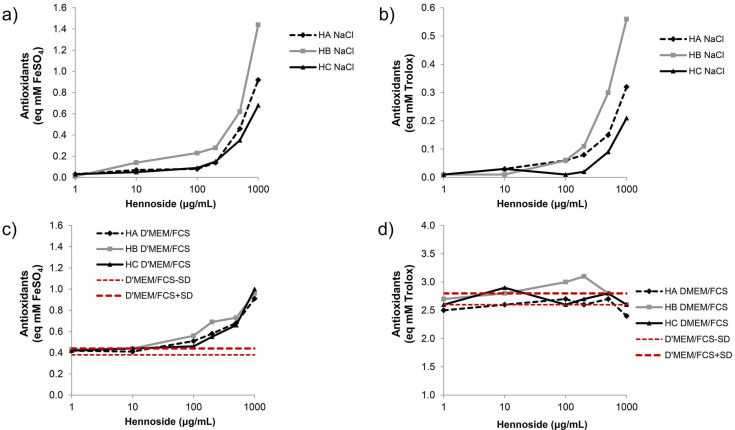
Antioxidant activity of hennosides in isotonic NaCl and Dulbecco’s Modified Eagle Medium supplemented with 10% fetal calf serum (D’MEM/FCS). Antioxidant activity measured by FRAP (**a**,**c**) and ABTS (**b**,**d**) assay. HA—hennoside A; HB—hennoside B; HC—hennoside C; NaCl—0.9% NaCl; D’MEM/FCS: D’MEM cell culture medium supplemented with 10%FCS; (**a**,**b**) Values represent the concentration of antioxidants diluted in 0.9% NaCl. (**c**,**d**) Values represent the concentration of antioxidants diluted in D’MEM + 10%FCS; Red lines indicate mean—1SD (D’MEM/FCS − SD) to mean + 1SD (D’MEM/FCS + SD) range of antioxidant concentration in D’MEM/FCS.

**Figure 3 plants-10-00237-f003:**
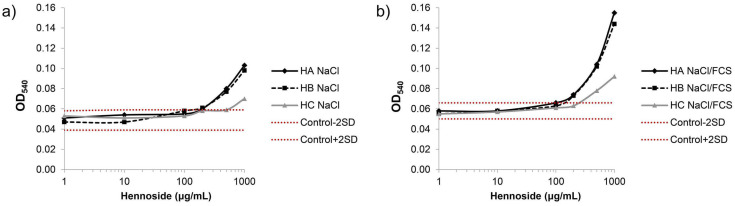
Hennoside-induced lysis of erythrocytes resuspended in (**a**) 0.9% NaCl and (**b**) 0.9% NaCl containing 10% FCS (NaCl/FCS). The hemolysis was followed based on the optical density at 540 nm (OD_540_) values of supernatants obtained after centrifugation of erythrocyte suspension. HA—hennoside A; HB—hennoside B; HC—hennoside C; NaCl—hennosides and erythrocytes diluted in 0.9% NaCl; NaCl/FCS—hennosides and erythrocytes diluted in 0.9% NaCl containing 10% FCS. Red lines indicate mean − 2SD (control − 2SD) to mean + 2SD (control + 2SD) range of control i.e., spontaneous lysis of erythrocytes incubated in 0.9% NaCl (**a**) and (**b**) 0.9% NaCl/FCS without hennosides.

**Figure 4 plants-10-00237-f004:**
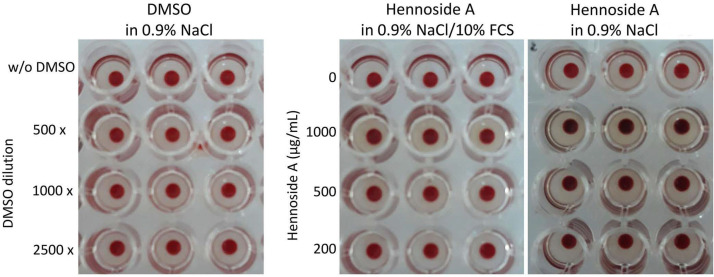
Hennoside-induced changes in color of erythrocytes. The image depicts the remaining non-lysed/intact erythrocytes after 90 min incubation, at 37 °C with hennosides diluted in 0.9% NaCl or 0.9% NaCl containing 10% FCS. DMSO diluted in 0.9% is used as a control. The results show non-hemolytic effect of DMSO.

**Figure 5 plants-10-00237-f005:**
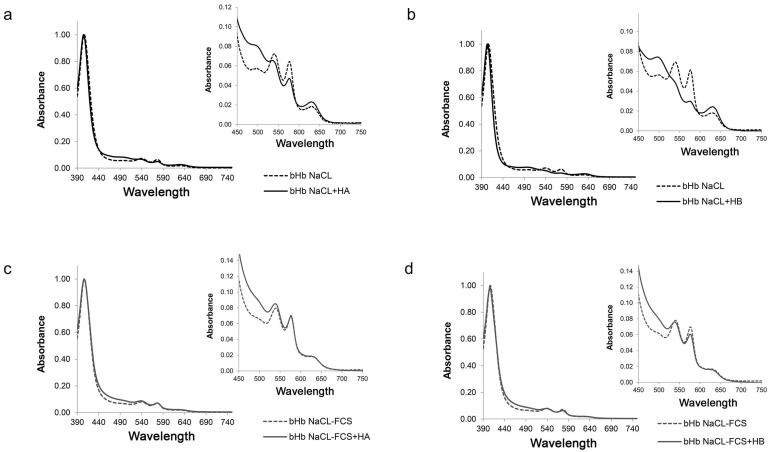
Hennoside-induced changes of Vis spectral characteristic (390–750 nm; zoom region 450–750 nm) of hemoglobin in solution. (**a**,**c**) 500 μg/mL of hennoside A; (**c**,**d**) 500 μg/mL of hennoside B; (**a**,**b**) diluent 0.9% NaCl (NaCl); (**c**,**d**) diluent 0.9% NaCl containing 10% FCS (NaCl-FCS).

**Figure 6 plants-10-00237-f006:**
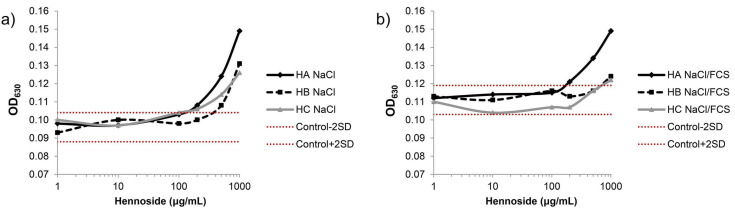
Hennoside-induced methemoglobin formation. Methemoglobin measurement based on an increase in OD_630_ values. (**a**) diluent 0.9% NaCl (NaCl); (**b**) diluent 0.9% NaCl containing 10% FCS (NaCl/FCS). Dotted red lines represent mean value ± 2SD range for the control, i.e., spontaneously formed methemoglobin.

**Figure 7 plants-10-00237-f007:**
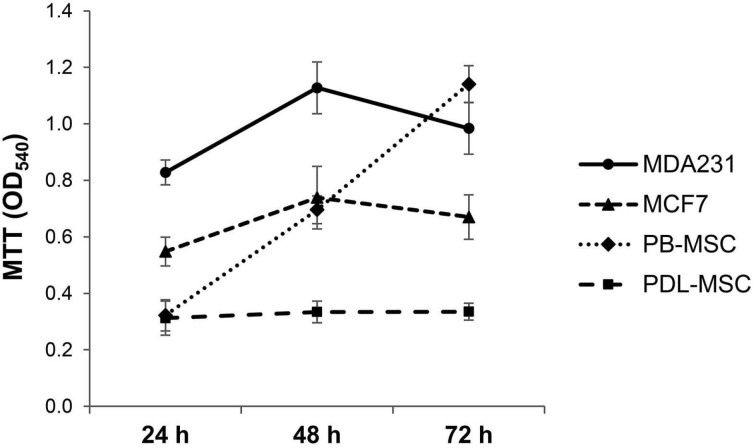
MTT test assayed spontaneous proliferation/viability of human breast cancer cells MDA 231 and MCF-7, and primary human periodontal ligament and peripheral blood mesenchymal stem cells (PDL-MSC and PB-MCS) in short-term cell culture.

**Figure 8 plants-10-00237-f008:**
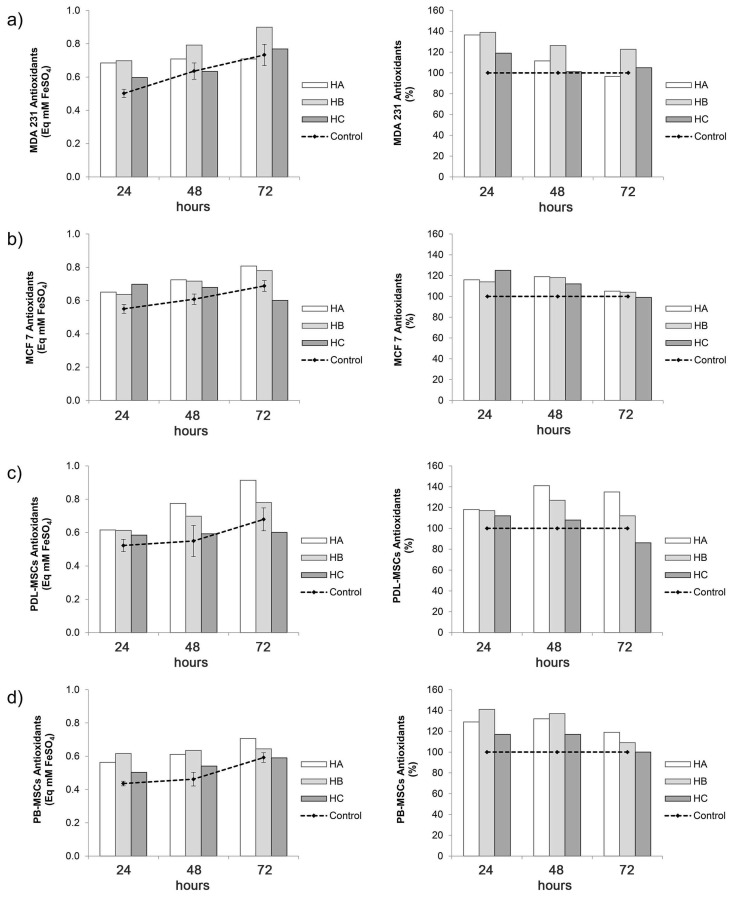
Hennosides at a concentration of 100 µg/mL enhance the concentration of antioxidants in supernatants of human breast cancer cells MDA 231 (**a**) and MCF-7 (**b**), and primary human periodontal ligament and peripheral blood mesenchymal stem cells, PDL-MSC (**c**) and PB-MCS (**d**) in short-term cell culture. The result of FRAP assay is given as the concentration of antioxidants presented as mM equivalent to FeSO4 concentration (mM), and as a percentage of corresponding control. Control (set at 100%) is the concentration of antioxidants in supernatants of cells grown without hennosides.

**Table 1 plants-10-00237-t001:** Effect of 500 μg/mL of hennosides A and B on Vis spectral characteristics of hemoglobin.

	Soret Band	Δα/Δβ	OD_Soret_/OD_575_	OD_630_
Hb in NaCl	406	0.70	15.6	0.018
Hb in NaCl + HA	405	0.29	20.4	0.026
Hb in NaCl-FCS	407	0.64	14.1	0.019
Hb in NaCl-FCS + HA	406	0.54	14.1	0.022
Hb in NaCl	406	0.69	15.9	0.020
Hb in NaCl + HB	403	0.00	30.3	0.028
Hb in NaCl-FCS	406	0.71	13.9	0.019
Hb in NaCl-FCS + HB	406	0.46	15.6	0.020

HA—hennoside A; HB—hennoside B; Δα/Δβ = (OD_577_–OD_560_)/(OD_541_–OD_560_); NaCl—0.9% NaCl; NaCl-FCS—0.9% NaCl supplemented with 10% FCS.

**Table 2 plants-10-00237-t002:** Viability of hennoside-treated cells: MTT test.

		Relative Viability (%)		
	Hennoside	24 h	48 h	72 h
	(μg/mL)			
		MDA 231		
HA	0	100	100	100
	100	116	101	107
HB	0	100	100	100
	100	101	95	104
HC	0	100	100	100
	100	109	95	105
		MCF 7		
HA	0	100	100	100
	100	118	117	118
HB	0	100	100	100
	100	116	111	114
HC	0	100	100	100
	100	91	90	116
		PDL-MSC		
HA	0	100	100	100
	100	98	99	88
HB	0	100	100	100
	100	104	107	92
HC	0	100	100	100
	100	95	99	83
		PB-MSC		
HA	0	100	100	100
	100	107	102	99
HB	0	100	100	100
	100	105	102	93
HC	0	100	100	100
	100	107	110	96

MTT reactivity of the cells grown in the cell culture medium without hennosides (control) was set at 100%. The relative viability of cells grown with hennosides is expressed as a percentage of corresponding control. Values are mean of two independent experiments. HA—hennoside A; HB—hennoside B; HC—hennoside C. MDA 231 and MCF 7—human breast cancer cell lines; PDL-MSC—human periodontal ligament mesenchymal stem cells; PB-MSC—human peripheral blood mesenchymal stem cells.

**Table 3 plants-10-00237-t003:** Hennosides: ^1^H NMR recorded data ^a^.

Compound	Hennoside A	Hennoside B	Hennoside C
H-5	8.00, d, J = 8.1 Hz	8.32, d, J = 8.1 Hz	7.68, d, J = 8.1 Hz
H-6	7.42, t, J = 8.1 Hz	6.96, t, J = 8.1 Hz	7.22, t, J = 8.1 Hz
H-7	7.22, t, J = 8.1 Hz	7.42, t, J = 8.1 Hz	7.24, t, J = 8.1 Hz
H-8	7.07, d, J = 8.1 Hz	7.24, d, J = 8.1 Hz	7.40, d, J = 8.1 Hz
H-3	6.48, s	5.87, s	6.18, s
H-1′	4.58, d, J = 6.5 Hz	4.72, d, J = 6.5 Hz	4.52, d, J = 6.5 Hz
H-2′-6′	3.80–3.26	3.80–3.26	3.80–3.26

^a^ NMR data recorded on a Varian Mercury instrument operating at 300 MHz (solvent CD3OD). The chemical shifts were expressed using the internal signal of CD2HOD (m5, 3.31 ppm) as reference.

## Data Availability

The datasets used and analyzed during the current study are available from the corresponding author on reasonable request.

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
