# Peer review of "Insight into the Biological Activity of Hennosides—Glucosides Isolated from Lawsonia inermis (henna): Could They Be Regarded as Active Constituents Instead"

_plants, 2021, doi:10.3390/plants10020237_

Round 1

Reviewer 1 Report

Dear Authors,

I have the following comments to your work:

  • please, remember to clarify the abbreviations at their first appearance in the text. at the moment going to results section directly from the introduction the reader is not sure about the actual meaning of the abbreviations used in the text
  • section 4.1.: add the time of plant material collection - the time of the year is especially important
  • line 254: 'extracted with methanol (100 ml) and sonicated' - where the leaves first extracted by maceration and later extracted again using ultrasound assisted extraction technique? if not, please, correct
  • section 4.2. how many milligrams of pure hennosides were obtained? which volume of the fractions was collected, which composition of the eluting solvent was used to recover given hennosides? how many mL of each eluting solvent did the authors use for the separation of these compounds?
  • what is DMEM, FCS, and others...please explain the used abbreviations
  • 4.3.1. and 4.3.2.: what was the concentration of the tested samples added to the reaction mixture? which range of absorbance value was obtained in the test?
  • 4.6. how about the normal cell lines...it is necessary to check whether these compounds are not cytotoxic against normal cells...
  • the abstract needs total re-arragement. it needs to contain facts and data elaborated in the designed study

Author Response

Answer to the Reviewer 1 questions and comments:

Reviewer 1: Please, remember to clarify the abbreviations at their first appearance in the text. At the moment going to results section directly from the introduction the reader is not sure about the actual meaning of the abbreviations used in the text.

Authors: All abbreviations have been clarified at their first appearance in the text.

Reviewer 1: Section 4.1.: add the time of plant material collection - the time of the year is especially important

Authors: The time of plant material collection has been added in the section 4.1. (revised manuscript, line 319-322.

Reviewer 1: Line 254: 'extracted with methanol (100 ml) and sonicated' - where the leaves first extracted by maceration and later extracted again using ultrasound assisted extraction technique? if not, please, correct:

Authors: In the revised manuscript, the methods of the sample preparation have been described with more details (line 327-333).

Reviewer 1: Section 4.2. how many milligrams of pure hennosides were obtained? which volume of the fractions was collected, which composition of the eluting solvent was used to recover given hennosides? how many mL of each eluting solvent did the authors use for the separation of these compounds?

Authors:  The requested data has been entered in the revised manuscript (line 333-340).

Reviewer 1: what is DMEM, FCS, and others...please explain the used abbreviations

Authors: All abbreviations have been clarified at their first appearance in the text.

Reviewer 1: 4.3.1. and 4.3.2.: what was the concentration of the tested samples added to the reaction mixture? which range of absorbance value was obtained in the test?

Authors: The concentration of tested samples and range of absorbance values for tested samples and standards were added in mentioned sections.

Reviewer 1: 4.6. how about the normal cell lines...it is necessary to check whether these compounds are not cytotoxic against normal cells...

Authors: An absence of cytotoxic effect of tested compound was confirmed based on the results of MTT assay shoving that the viability of primary mesenchymal stem cells was not decreased in the presence of any of hennosides isomers. It is pointed in the Section 2.4 (line 186-188 of the revised manuscript).

Reviewer 1: the abstract needs total re-arrangement. it needs to contain facts and data elaborated in the designed study

Authors: The abstract has been re-arranged according the Reviewer 1 and Reviewer 3 suggestion.

Additional comments:

Authors: Two references that were omitted by mistake in the original text have added in the revised manuscript:

  1. Putzbach et al., Anal Bioanal Chem 2007, 389, 197-205. doi: 10.1007/s00216-007-1409-6
  2. Jin et al., J Chromatogr A 2006, 1132, 320-324. doi: 10.1016/j.chroma.2006.08.022

Authors: A subsection “4.7. Antioxidant activity in the cell culture supernatants” has been added in the revised manuscript.

Reviewer 2 Report

Authors have worked on a very important topic, ie. medicinal effects of henna. However, it needs major revision based on my comments below. 

Describe FRAP and ABTS assay

Explain DMEM-FCS-/+SD in Figure 2c,d ? 

What does OD540 signify in Figure 3 ?

Explain Control +/-SD in Figure 3

Line 113/114, explain “reflected the change in hemoglobin molecules”

Line 114/115 Authors says, hennoside induced erythrocyte color change was not detected. However, I see the change in color at 1000ug/ml in same conditions in Figure 4

Any quantitative measurements on color change, which could be associated with hemolysis ? 

Line 123/124: Correct the English, “The effect was dependent on ”

Figure 5: Correct the spelling for Wavelength. Please clarify the zoom region on the right. Difference absorption spectra could be an easier way to present this data. The absorbance is below 0.1, is it really the signal or authors are interpreting the noise ? 

Line 124: Please explain why 390-750nm absorption spectrum is analyzed

Line 126-127: Please explain in it 

Figure 7: Since Authors are mentioning about 20% uncertainty in measurement tool, it is hard to say time-dependence in case of MCF-7 and MDA-231 cells. 

Explain results for Figure 8. 

Discussion, how chemistry was related to these results like cell viability, hemolysis and effects on hemoglobin. And how these effects are related to lawsone ? 

Check the sentence in line 197/198/199. Please make it clear. 

Line 203/204 check the grammer 

Author Response

Answer to the Reviewer 2 questions and comments

Authors have worked on a very important topic, ie. medicinal effects of henna. However, it needs major revision based on my comments below. 

Reviewer 2: Describe FRAP and ABTS assay

Authors: FRAP and ABTS assays have been described in detail in sections 4.3.1 and 4.3.2 sections of revised manuscript.

Reviewer 2: Explain DMEM-FCS-/+SD in Figure 2c,d? 

Authors: “DMEM-FCS-/+SD” in Figure 2 c,d has been explained in the Figure 2 legend.

Reviewer 2: What does OD540 signify in Figure 3?

Authors: “OD540” has been explained in Figure 3 legend.

Reviewer 2: Explain Control +/-SD in Figure 3.

Authors: “Control +/-SD” has been explained in the Figure 3 legend.

Reviewer 2: Line 113/114, explain “reflected the change in hemoglobin molecules”

Authors: Hennoside induced changes of hemoglobin molecules have been described in the Section 2.3. In the section 2.2 of the revised manuscript we have added a sentence “Some of hennoside induced hemoglobin structural changes have been described in in the section 2.3” (Revised manuscript, line 126/127).

Reviewer 2: Line 114/115. Authors says, hennoside induced erythrocyte color change was not detected. However, I see the change in color at 1000 ug/ml in same conditions in Figure 4.

Authors: Thank you for this comment. We have replaced the sentence “Conversely, when erythrocytes and hennosides were diluted in 0.9% NaCl /10% FCS, the hennoside induced erythrocyte color change was not detected” with the sentence “Conversely, when erythrocytes and hennosides were diluted in 0.9% NaCl /10% FCS, the hennoside induced erythrocyte color change was detected only at the 1000 μg/mL concentration”. (revised manuscript, line 128/129).

In this paragraph we have noticed a type error, and “≥ 500 μg/ml” have been replaced with "≥ 200 μg/ml”. (Revised manuscript, line 124)

Reviewer 2: Any quantitative measurements on color change, which could be associated with hemolysis ?

Authors: Data on spontaneous and hennoside induced hemolysis level are given in Figure 2 (manuscript body) and Table S2 (Supplementary material).

Reviewer 2: Line 123/124: Correct the English, “The effect was dependent on ”

Authors: “The effect was hennoside type and solvent type dependent” have been replaced with “The effect was dependent on hennoside isoform and solvent type”. (revised manuscript, line 137/138)

Reviewer 2: Figure 5: Correct the spelling for Wavelength. Please clarify the zoom region on the right. Difference absorption spectra could be an easier way to present this data. The absorbance is below 0.1, is it really the signal or authors are interpreting the noise?

Authors: Spelling for “wavelength” has been corrected. The absorbance below 0.1 is signal, not noise.

Reviewer 2: Line 124: Please explain why 390-750 nm absorption spectrum is analyzed

Reviewer 2: Line 126-127: Please explain in it

Authors: The more detailed explanation for Vis spectra analysis has been given in the Section 2.3, (line 137/151) of revised manuscript. In this text three new references have been quoted:

  1. Zijlstra et al., Clin Chem 1991, 37, 1633-1638.
  2. Sherif and Amal. Romanian Journal of Biophysics 2010, 20, 269-281.
  3. Hanson and Ballantyne, PLoS One 2010, 5(9):e12830.

Reviewer 2: Figure 7: Since Authors are mentioning about 20% uncertainty in measurement tool, it is hard to say time-dependence in case of MCF-7 and MDA-231 cells. 

Authors: Thank you for this comment. The text of time-dependence of the analyzed cells has growth has been changed in the revised manuscript (line 183/186). However, MDA-231 cells increased the MTT reactivity for up to 40% in 48 h culture, which mean that MTT test detect an increase in their viability.

Reviewer 2: Explain results for Figure 8. 

Authors: Results for Figure 8 have been explained in the Section 2.5 (line 205/223) of the revised manuscript.

Reviewer 2: Discussion, how chemistry was related to these results like cell viability, hemolysis and effects on hemoglobin. And how these effects are related to lawsone?

Authors: Answers to these questions we have included in the Discussion (revised manuscript, line 283-316).

To support this part of the Discussion we have additionally quoted five references:

  1. Zinkham et al, Pediatrics 1996, 97, 707-709.
  2. Das et al, International Research Journals 2020, 11, 1-7.
  3. Matulich and Sullivan, Contact Dermatitis 2005, 53, 33-36.
  4. Jovanovic and Slavkoviv-Jovanovic, J Dermatol 2009, 36, 63-65.
  5. Lamchahab et al. Arch Pediatr 2011, 18, 653-656.

Reviewer 2: Check the sentence in line 197/198/199. Please make it clear. 

Authors: The sentence “Each of the chemicals henna contains, in addition to the lawsone, a widely-used hair dye, has its own effect on the body to be considered” has been replaced with “Each of those chemicals that henna contains, in addition to well-known lawsone, a widely used hair dye, can exert its own biological effect on the body”. (Revised manuscript, line 235/236)

Reviewer 2: Line 203/204 check the grammer

Authors: The sentence “According to our knowledge there is lack of data of the effect of in vitro survival of erythrocytes exposed to hennosides, and we found of interest to explore the redox potential of these isomers on erythrocytes” has been replaced with “According to our knowledge there is lack of data on biological effects of hennosides, and we found of interest to explore the redox potential of these isomers using erythrocytes as a model system”. (Revised manuscript, line 241/243).

Additional comments:

Authors: Two references that were omitted by mistake in the original text have added in the revised manuscript:

  1. Putzbach et al., Anal Bioanal Chem 2007, 389, 197-205. doi: 10.1007/s00216-007-1409-6
  2. Jin et al., J Chromatogr A 2006, 1132, 320-324. doi: 10.1016/j.chroma.2006.08.022

Authors: A subsection “4.7. Antioxidant activity in the cell culture supernatants” has been added in the revised manuscript.

Reviewer 3 Report

Dear  authors, 

I appreciate your valuable research by presented manuscript. My notes directed to some technical aspects as follow:

Abstract need to be updated by including some essential findings from your results or to extend by including analytical methods presented in your work.

Unify the units of measurement as required (ml or mL)

Row 281: FeSO4x7H2O -> FeSO4x7H2O

Row 303: 0.9% NaCl or 0.9 ? NaCl

Row 336: 1×104 -> 1x104

Conclusions are present quite general as well as not supported by results. It would be good to support your conclusions with key points from the achieved results.

Author Response

Answer to the Reviewer 3 questions and comments

Dear authors,

I appreciate your valuable research by presented manuscript. My notes directed to some technical aspects as follow:

Reviewer 3. Abstract need to be updated by including some essential findings from your results or to extend by including analytical methods presented in your work.

Authors: The abstract has been re-arranged according suggestion of Reviewer 1 and Reviewer 3.

Reviewer 3. Unify the units of measurement as required (ml or mL)

Authors: The units of measurement are unified, and “ml” has been replaced with “mL” throughout the text.

Reviewer 3. Row 281: FeSO4x7H2O -> FeSO4x7H2O

Authors: “FeSO4x7H2O” has been replaced with “FeSO4x7H2O” (revised manuscript, row 365).

Reviewer 3. Row 303: 0.9% NaCl or 0.9 ? NaCl

Authors: “0.9 NaCl” has been replaced with “0.9% NaCl” (revised manuscript, row 388).

Reviewer 3. Row 336: 1×104 -> 1x104

Authors: “1×104” has been replaced with “1x104” (revised manuscript, row 377).

Reviewer 3. Conclusions are present quite general as well as not supported by results. It would be good to support your conclusions with key points from the achieved results.

Authors: Conclusions have been changed according to Reviewer’s demand.

Additional comments:

Authors: Two references that were omitted by mistake in the original text have added in the revised manuscript:

  1. Putzbach et al., Anal Bioanal Chem 2007, 389, 197-205. doi: 10.1007/s00216-007-1409-6
  2. Jin et al., J Chromatogr A 2006, 1132, 320-324. doi: 10.1016/j.chroma.2006.08.022

Authors: A subsection “4.7. Antioxidant activity in the cell culture supernatants” has been added in the revised manuscript.

Round 2

Reviewer 1 Report

Dear Authors, thank you very much indeed for the introduced corrections.

I would like to ask you to focus on the abstract of the manuscript.

Please, go through the text again and remove the repetitions (utilization - 2nd and 3rd line, plant - line 2 and 4, etc.)

Please, try to make the introduction more smooth.

Line 30, please add to the sentence: "in the cell culture supernatants at the concentration of....."

line 27 - also add tot he sentence the obtained values

please, think the content over. first the Authors write about the prooxidative activity and later about the antioxidant potential.  ot to confuse the readers it would be good to emphasize the difference in te concentration and indicate that these properties are beneficial. 

add the botanical family to the keywords and abstract.

Author Response

Reviewer: Dear Authors, thank you very much indeed for the introduced corrections.

I would like to ask you to focus on the abstract of the manuscript.

Please, go through the text again and remove the repetitions (utilization - 2nd and 3rd line, plant - line 2 and 4, etc.)

Please, try to make the introduction more smooth.

Line 30, please add to the sentence: "in the cell culture supernatants at the concentration of....."

line 27 - also add to the sentence the obtained values

Authors: The abstract has been rearranged according to the Reviewers suggestions.

Reviever: please, think the content over. first the Authors write about the prooxidative activity and later about the antioxidant potential.  ot to confuse the readers it would be good to emphasize the difference in the concentration and indicate that these properties are beneficial. 

Authors: We would like to point out that by using experimental techniques described in the section Material and Methods, it has been shown that hennoside isomers at the concentration of 500 and 1000 μg/ml showed both pro- and anti-inflammatory activity (the results that were shown in Figures 2-6, Table 1, and Tables S1-S3 in the Supplementary material).

In the revised Discussion of manuscript (line 315-329) we have written that our results only showed that hennosides are redox active molecules and that “Whether hennoside isomers, as potential therapeutics, will act either as pro- or antioxidant agents depends on their concentration, the molecular substrates they react with, nutritional status and immunological competence of individuals as well as inflammatory milieu in organs and tissues affected by the disease “. Therefore, it is necessary to proceed with further investigation to determinate the mechanisms of action of hennosides …”  In this part of text two additional references have been quoted:

  1. Dharmaraja J Med Chem 2017, 60, 3221-3240.
  2. Bar-Or et al. Redox biol, 2015, 4, 340-345.

We are aware that with this study “we have just scratched the surface of hennosides biological activity”, and that additional experiments are needed to discuss the beneficial effects of hennosides in more details.

Reviewer: Add the botanical family to the keywords and abstract.

Authors: Botanical family have been added to the keywords and abstract

Authors’ additional comments:

  1. Lines 335-337: The time of plant material collection has been explained more properly.
  2. Minor spelling and type errors have been corrected.

Reviewer 2 Report

Authors have worked on most of my comments. 

Author Response

Dear Reviever,

Thank you.

Faithfully,

Authors